# communications
# engineering

# Deep learning-based optical coherence tomography angiography image construction using spatial vascular connectivity network

David Le [1], Taeyoon Son[1], Tae-Hoon Kim [1], Tobiloba Adejumo [1], Mansour Abtahi [1], Shaiban Ahmed[1], Alfa Rossi [1], Behrouz Ebrahimi [1], Albert Dadzie[1], Guangying Ma[1], Jennifer I. Lim[2] & Xincheng Yao [1,2 ✉]

Optical coherence tomography angiography (OCTA) provides unrivaled capability for depth-resolved visualization of retinal vasculature at the microcapillary level resolution. For OCTA image construction, repeated OCT scans from one location are required to identify blood vessels with active blood flow. The requirement for multi-scan-volumetric OCT can reduce OCTA imaging speed, which will induce eye movements and limit the image field-of-view. In principle, the blood flow should also affect the reflectance brightness profile along the vessel direction in a single-scan-volumetric OCT. Here we report a spatial vascular connectivity network (SVC-Net) for deep learning OCTA construction from single-scan-volumetric OCT. We quantitatively determine the optimal number of neighboring B-scans as image input, we compare the effects of neighboring B-scans to single B-scan input models, and we explore different loss functions for optimization of SVC-Net. This approach can improve the clinical implementation of OCTA by improving transverse image resolution or increasing the field-of-view.

[1] Department of Biomedical Engineering, University of Illinois at Chicago, Chicago, IL 60607, USA. [2] Department of Ophthalmology and Visual Sciences, University of Illinois at Chicago, Chicago, IL 60612, USA. ✉email: xcy@uic.edu

Optical coherence tomography (OCT) enables the non-invasive visualization of individual retinal layers with micrometer-level resolution. As one modality extension of OCT, OCT-angiography (OCTA) provides unparalleled capability for depth-resolved visualization of retinal vasculature at the microcapillary level resolution. OCTA is label-free and thus is completely non-invasive, compared to traditional fluorescein angiography. Studies have shown that OCTA provides improved capability for detecting subtle vascular distortions associated with the progression of retinal pathology, such as vessel dropout, foveal abnormalities, and increased vessel tortuosity[1,2]. Studies demonstrate that OCTA could even detect microaneurysms that were undetected on dilated clinical examination[3].

The principle of OCTA is that repeated OCT scans from one location are acquired for temporal vascular connectivity (TVC) processing to map retinal vasculature at the microcapillary resolution. Therefore, OCTA can be obtained from existing OCT systems with the addition of unique scan protocols and data processing algorithms[4]. However, the fundamental similarity between all OCTA instruments is that repeated OCT scans from one location are required for correlation analysis of sequential images to identify regions with active blood flow. Therefore, OCTA requires higher imaging speeds than most currently available OCT systems can provide in order to obtain a densely sampled volume. Conventional OCT device scanning speeds would result in too much trade-off between decreased field of view, lower image quality, and greatly increased scanning time. Additionally, the prolonged scanning time may also increase the potential effect of motion artifacts, such as blinking and microsaccades [5].

A potential solution lies in the use of deep learning algorithms. In recent years, deep learning, a subset of machine learning and artificial intelligence, has been making strides in ophthalmic research[6–10]. The principle behind deep learning is that the algorithm can learn directly from the training data and can objectively perform the required task. An example application is deep learning for artificial intelligence screening of retinopathies[11–15]. Current screening procedures require clinicians to manually examine retinal photographs. This can, therefore, lead to inter- and intra-rater variability; the same clinician could classify the same image differently on different days. Furthermore, to manually screen retinal photographs is a time-consuming process. Therefore, the deployment of artificial intelligence algorithms could alleviate these problems. Recent studies in deep learning OCTA have primarily been focused on the classification of eye diseases such as diabetic retinopathy[16–18], age-related macular degeneration[19–21], and glaucoma[22–24]. Other applications include improving the image quality of OCTA[25,26] and artery–vein segmentation[27–31]. Recently, deep learning has also been explored for OCTA construction[32–35]. While deep learning algorithms can detect large blood vessel branches in OCT readily, it is technically challenging to identify microcapillaries reliably.

We hypothesize that deep learning could leverage spatial vascular connectivity (SVC), i.e., brightness connectivity along the vessel direction, in a single-scan-volumetric OCT for OCTA construction. In this study, we train a convolutional neural network, titled SVC-Net, that leverages SVC inputs for OCTA construction. We show that SVC can be used reliably to predict microcapillary structures. We verify the feasibility of a deep learning approach using a dataset composed of single-scan-volumetric OCTs from animal and human eyes. In addition, we compare the differences between TVC and SVC-based signals in traditional OCTA construction and perform ablation studies on the optimization of deep learning models by different loss functions.

## Results

**The deep learning framework**. In this section, we provide an overview of our deep learning-based method for OCTA construction from a single-scan OCT volume. A conceptual diagram of our proposed methodology is shown in Fig. 1a. Our hypothesis is that the blood flow should affect the reflectance brightness profile along the vessel direction (Fig. 1b). In other words, spatial intensity variance among the vessels in a single-scan-volumetric OCT, as shown in Fig. 1c, can be equivalent to the temporal intensity variance of the same vessel location in the sequential images in the multi-scan-volumetric OCT for conventional OCTA construction. Our deep learning-based OCTA construction framework, which we term SVC-Net (for details of network architecture, see "Methods" section), was trained and tested using spectral domain-OCT images acquired from mouse and human retina. The ground truth is based on conventional speckle variance OCTA construction from four repeated OCT B-scans. The

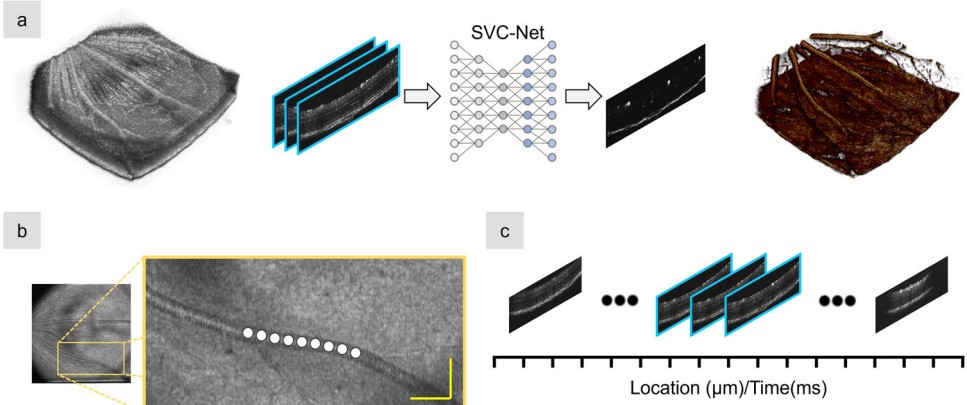

**Fig. 1 Illustration of the deep learning framework for OCTA construction from single-scan OCT volume.** Illustration of the concept and signal source of spatial vascular connectivity (SVC) are shown. **a** An illustration of the deep learning pipeline for OCTA construction using an SVC network (SVC-Net). The input is derived from a singular OCT volume and is comprised of three neighboring OCT B-scans outlined in blue. The output is a single OCTA B-scan. In the training process, the ground truth is derived from conventional OCTA construction, i.e., speckle variance. **b** An illustration of the vascular spatial connectivity information found in the OCT B-scans. The white circles highlight how retinal vessels are connected via the neighboring OCT B-scans. **c** A representation of the spatial and temporal information in neighboring B-scans, outlined in blue, in a single OCT volume. The yellow scale bar in **b** represents 100 μm.

input into SVC-Net will be comprised of OCT B-scans from single-scan OCT volume, and the output of SVC-Net will be the OCTA B-scan.

**Optimization of neighboring scans**. To leverage deep learning as a potential strategy for reliable OCTA construction, we first determine if SVC can provide the required information that the model can learn from. Therefore, we performed an ablation study to evaluate the effects of the different numbers of B-scans used for OCTA construction using the TVC, i.e., repeated B-scans, and the SVC, i.e., adjacent neighbor B-scans. To compare TVC and SVC, we use intensity-based speckle variance to generate OCTA. This procedure will help to determine the optimal number of neighboring B-scans to be used as input into SVC-Net. We illustrate the effect of the number of B-scans used on OCTA image quality using TVC and SVC. In our dataset, each volume contains four repeated B-scans. For qualitative comparison, we calculate OCTA for different numbers of B-scans, namely two, three, and four, which we refer to as 2 N, 3 N, and 4 N, respectively. The results of this ablation study are illustrated in Fig. 2 and Fig. 3 for animal and human datasets, respectively.

For TVC-based speckle variance (SV) processing, visual observations show as the number of repeated B-scans increases, the noise is reduced, as illustrated by Fig. 2 and Fig. 3. Therefore, the four B-scans for TVC-based SV processing have the best image quality. On the other hand, for SVC, there is a different trend; it can be observed that there is an optimal number of B-scans for SVC-based SV processing. For the animal dataset, we can observe that the SVC-2N has the worst image quality and that the SVC-3N and SVC-4N have comparable image quality. However, in the human dataset, SVC-4N, as compared to SVC-3N, decreases the vascular detail due to the increase in a blur effect, which can be visibly observed in the representative en face. This type of artifact is commonly described in OCTA as 'vessel doubling' due to poor registration. In the case of SVC-4N, since the vessels are not completely duplicated, we can refer to this artifact as pseudo-vessel doubling.

For the quantitative comparison, since the TVC-4N has the best qualitative performance, it will be the ground truth for comparison. We quantify the multi-scale structural similarity index measure (MS-SSIM) and the peak-signal-to-noise-ratio (PSNR) for the TVC 2 N and 3 N, and the SVC 2 N, 3 N, and 4 N en face images. The statistical information of this analysis is summarized in Tables 1 and 2. For the animal dataset, we confirm our qualitative observations that increasing the number of B-scans improves performance, as the TVC-3N has the best overall MS-SSIM and PSNR. Meanwhile, for the SVC, we observe that for the animal dataset, there is a decreasing trend for the MS-SSIM from SVC-2N to SVC-4N. However, we observed for the PSNR, the optimal number of B-scans was the SVC-3N. For the human dataset, we observe similar trends as compared to the mouse dataset, with the TVC having an improved performance on both metrics when using more B-scans. For the SVC metrics, we observe that for both the MS-SSIM and PSNR, the optimal number of B-scans was using SVC-3N since it has better performance than SVC-2N and SVC-4N. Therefore, for SVC-Net, based on qualitative observation and quantitative analysis, we will use a 3 N input, i.e., comprised of three adjacent neighboring B-scans.

**Microcapillary vessels visualization**. The primary usage of OCTA is to observe the en face projections of the retinal vascular layers. Therefore, we perform both qualitative and quantitative analyses for the en face projection of the superficial vascular plexus (SVP) and deep vascular plexus (DVP). To compare the effects of the SVC, we qualitatively compare the 1 N and 3 N models to the ground truth en face of the SVP and DVP for the animal eye in Fig. 4. For the SVP, we observe that large vessels were constructed in both the 1 N and 3 N models. However, an example of a large vessel that progressed to a smaller vessel was observed to have poor construction in the 1 N model. Whereas in the 3 N model, the same vessel was constructed properly. Showing that SVC helped preserve the details of smaller vessels. For the DVP, we observed that the 1 N model was able to predict

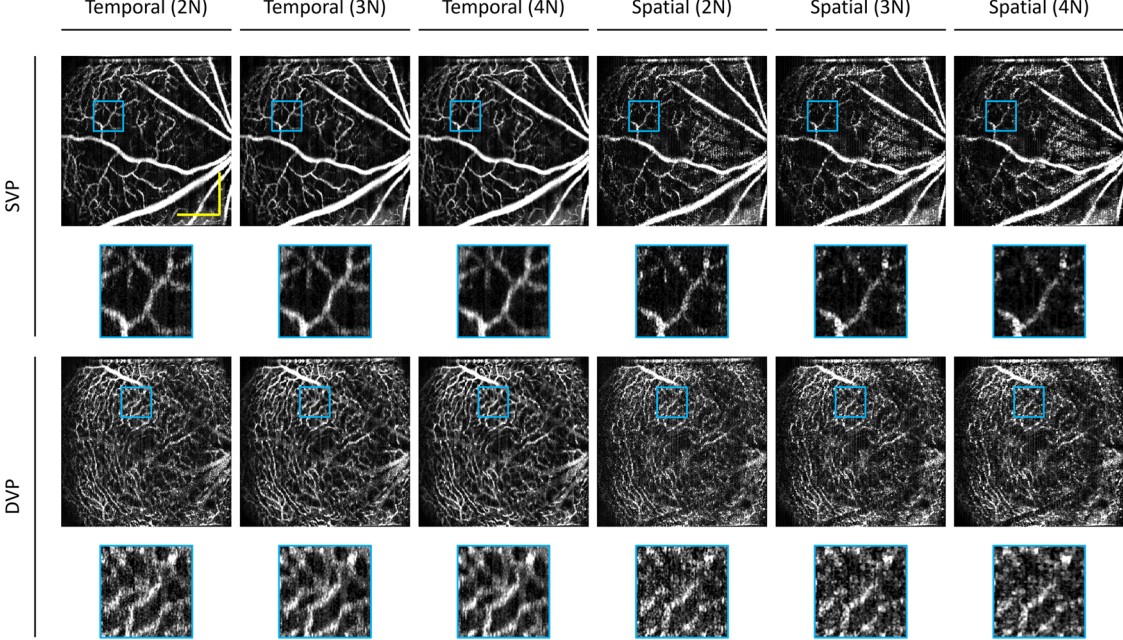

**Fig. 2 Comparison of conventional speckle variance OCTA construction in the mouse eye.** The temporal vascular connectivity (TVC) and spatial vascular connectivity (SVC) for varying numbers of B-scans in speckle variance processing are shown. Representative OCTA en faces for temporal and spatial approaches for superficial vascular plexus (SVP) and deep vascular plexus (DVP) layers. Zoomed regions of the images are outlined in blue. The yellow scale bar represents 300 µm.

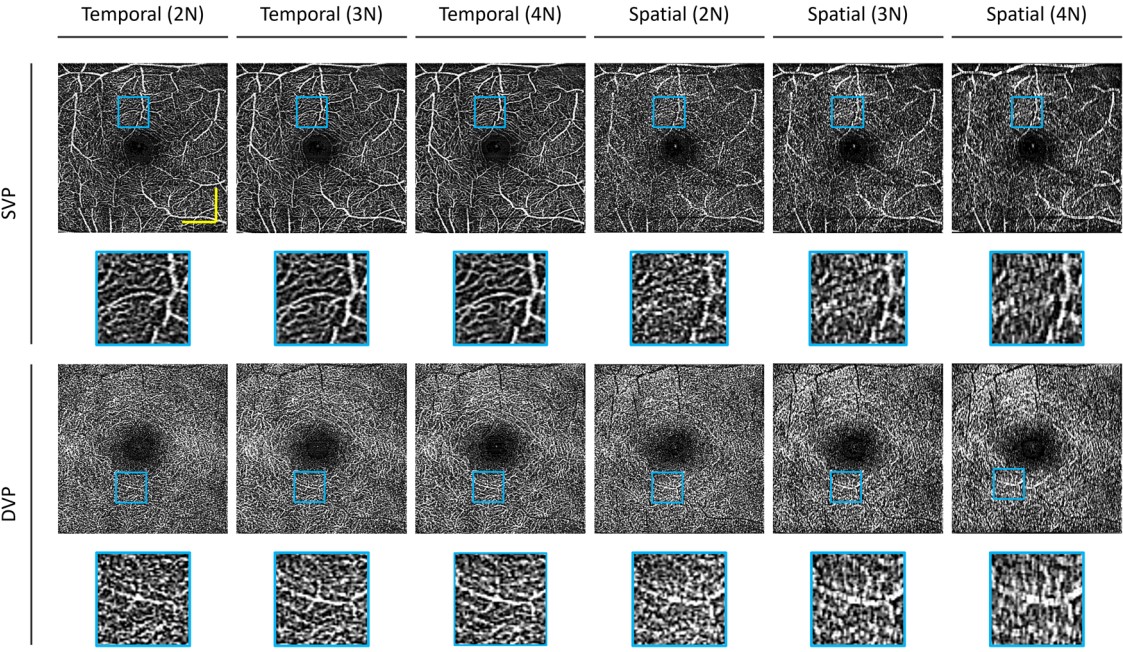

**Fig. 3 Comparison of conventional speckle variance OCTA construction in the human eye.** The temporal vascular connectivity (TVC) and spatial vascular connectivity (SVC) for varying numbers of B-scans in speckle variance processing are shown. Representative OCTA en faces for temporal and spatial approaches for superficial vascular plexus (SVP) and deep vascular plexus (DVP) layers. Zoomed regions of the images are outlined in blue. The yellow scale bar represents 600 μm.

**Table 1 Evaluation metrics on conventional speckle variance OCTA using temporal and spatial scans.**

| Dataset | Metric | Layer | Temporal 3 N (I) | Temporal 2 N (II) | Spatial 2 N (III) | Spatial 3 N (IV) | Spatial 4 N (V) | ANOVA |
|---|---|---|---|---|---|---|---|---|
| Mouse ($n = 6$) | MS-SSIM | SVP | 0.9728 ± 0.0070 | 0.934 ± 0.0093 | 0.8291 ± 0.0213 | 0.8193 ± 0.0286 | 0.7938 ± 0.0322 | <0.001 |
| | | DVP | 0.9487 ± 0.0278 | 0.8696 ± 0.0460 | 0.7422 ± 0.0430 | 0.7526 ± 0.0337 | 0.7285 ± 0.0268 | <0.001 |
| | PSNR | SVP | 30.03 ± 0.85 | 25.06 ± 0.77 | 20.25 ± 1.07 | 20.42 ± 1.23 | 19.84 ± 1.27 | <0.001 |
| | | DVP | 24.46 ± 2.06 | 19.71 ± 1.93 | 16.52 ± 1.79 | 17.16 ± 1.84 | 16.91 ± 1.86 | <0.001 |
| Human ($n = 6$) | MS-SSIM | SVP | 0.9467 ± 0.0117 | 0.8737 ± 0.0244 | 0.6251 ± 0.0446 | 0.6658 ± 0.0426 | 0.6318 ± 0.0407 | <0.001 |
| | | DVP | 0.8719 ± 0.0161 | 0.7295 ± 0.0270 | 0.546 ± 0.0270 | 0.5811 ± 0.0303 | 0.5356 ± 0.0271 | <0.001 |
| | PSNR | SVP | 18.22 ± 0.73 | 14.68 ± 0.67 | 10.33 ± 0.93 | 10.84 ± 0.98 | 10.53 ± 1.00 | <0.001 |
| | | DVP | 13.82 ± 0.35 | 10.82 ± 0.26 | 8.41 ± 0.21 | 8.67 ± 0.24 | 8.36 ± 0.20 | <0.001 |

Values are reported as mean ± standard deviation.
One-way ANOVA was performed for multi-group comparisons.

**Table 2 Post hoc analysis of the different methods of conventional speckle variance OCTA.**

| Dataset | Metric | Layer | I vs II | I vs III | I vs IV | I vs V | II vs III | II vs IV | II vs V | III vs IV | III vs V | IV vs V |
|---|---|---|---|---|---|---|---|---|---|---|---|---|
| Mouse ($n = 6$) | MS-SSIM | SVP | <0.001 | <0.001 | <0.001 | <0.001 | <0.001 | <0.001 | <0.001 | 0.075 | <0.01 | <0.001 |
| | | DVP | <0.001 | <0.001 | <0.001 | <0.001 | <0.001 | <0.001 | <0.001 | 0.065 | 0.198 | <0.01 |
| | PSNR | SVP | <0.001 | <0.001 | <0.001 | <0.001 | <0.001 | <0.001 | <0.001 | 0.112 | 0.017 | <0.001 |
| | | DVP | <0.001 | <0.001 | <0.001 | <0.001 | <0.001 | <0.001 | <0.001 | 0.065 | 0.198 | <0.01 |
| Human ($n = 6$) | MS-SSIM | SVP | <0.001 | <0.001 | <0.001 | <0.001 | <0.001 | <0.001 | <0.001 | <0.001 | 0.113 | <0.001 |
| | | DVP | <0.001 | <0.001 | <0.001 | <0.001 | <0.001 | <0.001 | <0.001 | <0.001 | <0.01 | <0.001 |
| | PSNR | SVP | <0.001 | <0.001 | <0.001 | <0.001 | <0.001 | <0.001 | <0.001 | <0.001 | <0.01 | <0.001 |
| | | DVP | <0.001 | <0.001 | <0.001 | <0.001 | <0.001 | <0.001 | <0.001 | <0.001 | <0.01 | <0.001 |

For individual comparisons, a pair-wise two-way Student's $t$-test was performed. I, Temporal 3N; II, Temporal 2N; III, Spatial 2N; IV, Spatial 3N; V, Spatial 4N

some capillary structures. However, due to the poor contrast, it has a noisier appearance. When compared to the 3 N model, the capillaries are reconstructed in finer detail.

Example visualization of the deep learning results on the effect of SVC on human eyes are shown in Fig. 5. On the en face OCT of both the SVP and DVP, as shown in Fig. 5, we can observe that

there are distinct capillary structures with less contrast as compared to OCTA. Therefore, it explains why we can observe that the 1 N and 3 N models are able to predict both the large and small vessel structures in the SVP, as both the intensity and structural information are present for the model to learn from in order to predict the vessels. In the DVP, we can observe that the

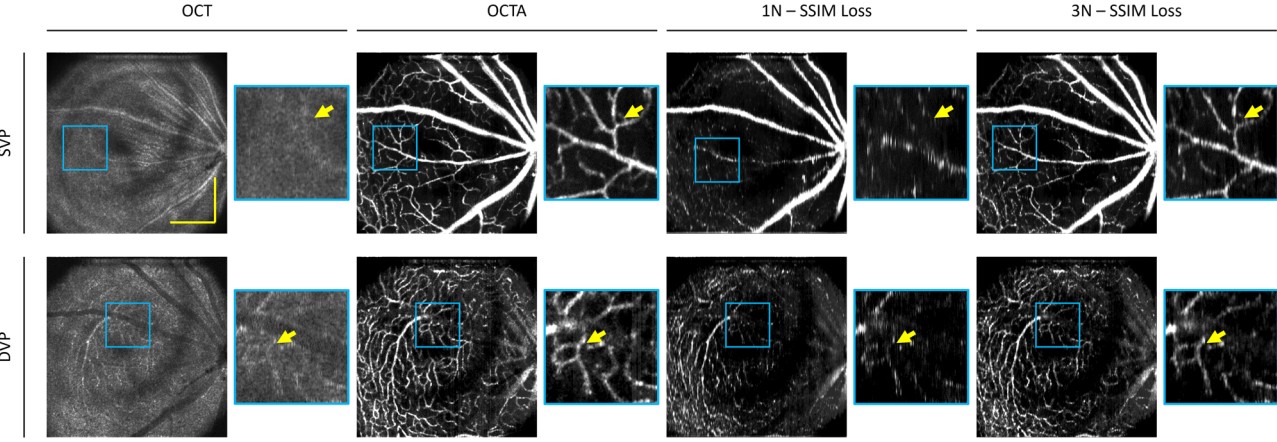

**Fig. 4 Effect of SVC on en face OCTA prediction on mouse eye.** Representative en face images from OCT, OCTA ground truth, and predictions using structural similarity index measure (SSIM) as the loss function for single scan (1 N) and three adjacent neighbors (3 N) inputs for superficial vascular plexus (SVP) and deep vascular plexus (DVP) layers. Zoomed regions of the images are outlined in blue. The yellow scale bar represents 300 μm.

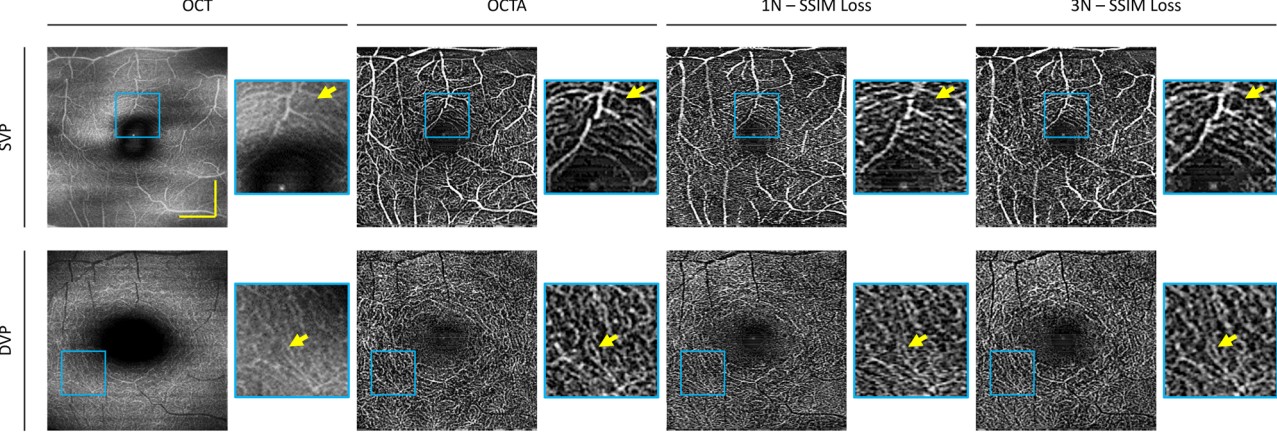

**Fig. 5 Effect of SVC on en face OCTA prediction on the human eye.** Representative en face images from OCT, OCTA ground truth, and predictions using structural similarity index measure (SSIM) as the loss function for single scan (1 N) and three adjacent neighbors (3 N) inputs for superficial vascular plexus (SVP) and deep vascular plexus (DVP) layers. Zoomed regions of the images are outlined in blue. The yellow scale bar represents 600 μm.

contrast in the 3 N model is better than the prediction of the 1 N model. This suggests that the SVC improves the model's performance to predict the finer structural details in capillary-level vessels.

**Effect of the loss function**. In this study, we also evaluate the effect of the different loss functions, i.e., mean squared error (MSE) and structural similarity index measure (SSIM) loss functions, for OCTA construction on the 1 N and 3 N models. Examples of output en face images from an animal eye of each model are illustrated in Fig. 6. Comparison between the 1 N models trained with MSE and SSIM, we observe a discernable increase in noise when the model is trained with the MSE loss function. In comparison to the 3 N models trained with MSE and SSIM, it results in noise reduction and contrast improvement, with the 3 N model trained with SSIM having the overall best contrast. In the DVP, we can observe that the 1 N trained with SSIM has a better capillary level structure as compared to the 1 N trained with the MSE, and when SVC is employed, it improves the contrast of the capillaries.

Next, we compare the effects of the loss functions on the model's performance in a human eye, as illustrated in Fig. 7. We observe that for the 1 N models, the model trained with the MSE loss function has higher levels of noise and poorer

contrast as compared to the 1 N model trained with the SSIM loss function. The modification of the loss function improved the 1 N's performance to produce the capillary level structures in higher contrast. Similar observations can be seen for the 3 N models. In the model trained with the MSE loss function, in the SVP, we can observe that some of the capillary level structures are predicted. However, there is relatively more noise. Whereas in the model trained with the SSIM loss function, the noise level is reduced. We can also observe that in the DVP, the capillary level structures seem more dilated; this could be due to the lower levels of contrast between the fine vessels. In the human dataset, the 3 N input with SSIM is the best-performing model and can produce vessel structures with higher contrast.

The evaluation metrics, MS-SSIM and PSNR, were quantified on both the SVP and DVP en faces to quantitatively compare the performances of the four models, and statistical analysis is summarized in Tables 3 and 4, respectively. For the MS-SSIM metric, it can be observed that, for both the animal and human datasets, the 1 N model trained with MSE had the lowest performance, followed by the 1 N model trained with the SSIM loss function, which had a slight improvement. The introduction of the SVC significantly improved the similarity between ground truth and predicted en face images. The 3 N model trained with MSE had significantly better results than both 1 N models.

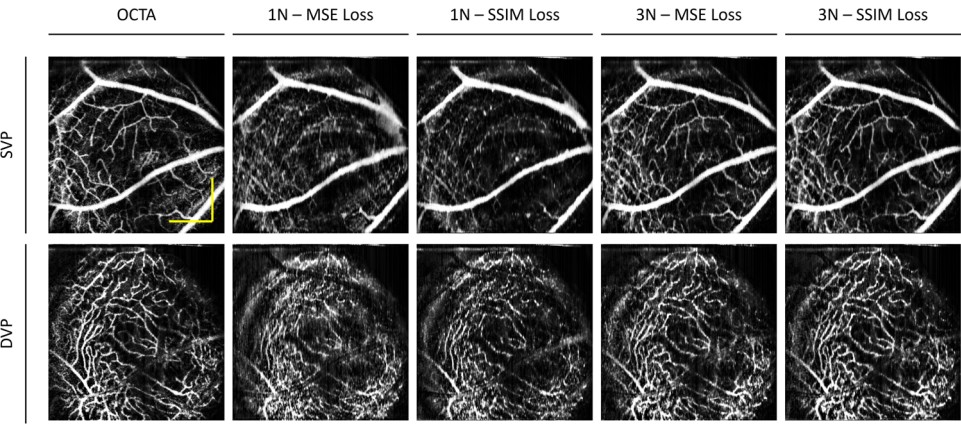

**Fig. 6 Effect of loss function on en face OCTA prediction on mouse eye.** Representative en face images from OCTA ground truth, and predictions using different loss functions, mean squared error (MSE) and structural similarity index measure (SSIM), and input images, single scan (1 N) and three adjacent neighbors (3 N), for superficial vascular plexus (SVP) and deep vascular plexus (DVP) layers. The yellow scale bar represents 300 μm.

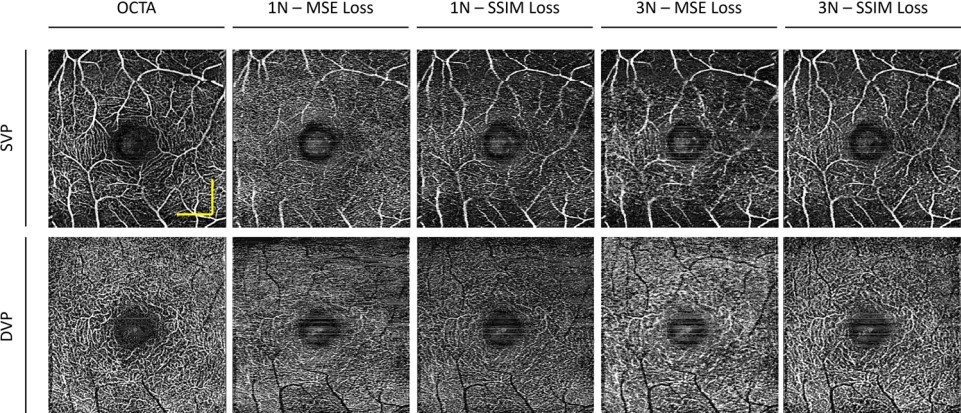

**Fig. 7 Effect of loss function on en face OCTA prediction on the human eye.** Representative en face images from OCTA ground truth, and predictions using different loss functions, mean squared error (MSE) and structural similarity index measure (SSIM), and input images, single scan (1 N) and three adjacent neighbors (3 N), for superficial vascular plexus (SVP) and deep vascular plexus (DVP) layers. The yellow scale bar represents 600 μm.

**Table 3 Evaluation metrics on deep learning results for different combinations of input type and loss function.**

| Dataset | Metric | Layer | 1 N-MSE (I) | 1 N-SSIM (II) | 3 N-MSE (III) | 3 N-SSIM (IV) | ANOVA |
|---|---|---|---|---|---|---|---|
| Mouse (n = 6) | MS-SSIM | SVP | 0.794 ± 0.061 | 0.806 ± 0.055 | 0.881 ± 0.039 | 0.901 ± 0.034 | 0.002 |
| | | DVP | 0.752 ± 0.078 | 0.785 ± 0.061 | 0.842 ± 0.054 | 0.860 ± 0.046 | 0.021 |
| | PSNR | SVP | 20.78 ± 3.09 | 20.84 ± 2.88 | 23.16 ± 3.13 | 23.99 ± 2.76 | 0.178 |
| | | DVP | 23.18 ± 4.27 | 23.80 ± 4.04 | 25.26 ± 4.16 | 25.85 ± 3.91 | 0.654 |
| Human (n = 6) | MS-SSIM | SVP | 0.641 ± 0.062 | 0.706 ± 0.047 | 0.733 ± 0.048 | 0.760 ± 0.044 | 0.004 |
| | | DVP | 0.553 ± 0.056 | 0.583 ± 0.046 | 0.627 ± 0.048 | 0.669 ± 0.036 | 0.002 |
| | PSNR | SVP | 14.48 ± 1.41 | 15.31 ± 1.41 | 16.65 ± 1.62 | 16.73 ± 1.61 | 0.049 |
| | | DVP | 12.80 ± 0.62 | 13.24 ± 0.44 | 13.73 ± 0.87 | 14.25 ± 0.56 | 0.005 |

Values are reported as mean ± standard deviation.
One-way ANOVA was performed for multi-group comparisons.

With the modification of the loss function, the 3 N model trained with SSIM had the best performance.

For the PSNR measurement, we observed that in the SVP, there were only slight differences between the loss functions while using the same input types (single or neighbored inputs). This may be due to the presence of large vessels in the SVP, which, regardless of the loss functions, the convolutional neural network (CNN) was able to predict consistently and had maximum pixel intensity, e.g., 255. However, when observing the quantitative evaluations for the DVP, we observe PSNR distinguishable improvements of the loss function and input type on the model's performance. This may be due to the abundance of smaller capillary level structures, where if the predicted image had an increase in noise, it could be reflected in the PSNR value. The use of SVC and the SSIM loss function reduced the noise level and improved the PSNR. For both the mouse and human datasets, the 3 N model trained with the SSIM loss function had the best overall performance.

**Table 4 Post hoc analysis of the different deep learning models as compared to the ground truth conventional OCTA.**

| Dataset | Metric | Layer | I vs II | I vs III | I vs IV | II vs III | II vs IV | III vs IV |
|---|---|---|---|---|---|---|---|---|
| Mouse (n = 6) | MS-SSIM | SVP | 0.739 | <0.01 | <0.01 | <0.01 | <0.01 | 0.042 |
| | | DVP | 0.012 | <0.01 | <0.01 | 0.01 | <0.01 | 0.056 |
| | PSNR | SVP | 0.744 | <0.01 | <0.01 | <0.01 | <0.01 | 0.141 |
| | | DVP | <0.01 | <0.01 | <0.01 | 0.026 | 0.016 | 0.02 |
| Human (n = 6) | MS-SSIM | SVP | 0.068 | <0.01 | <0.01 | <0.01 | <0.01 | 0.017 |
| | | DVP | 0.069 | <0.01 | <0.01 | <0.01 | <0.01 | <0.01 |
| | PSNR | SVP | <0.01 | <0.01 | <0.01 | <0.01 | <0.01 | 0.583 |
| | | DVP | 0.131 | <0.01 | <0.01 | 0.209 | <0.01 | 0.011 |

For individual comparisons, a pair-wise two-way Student's *t*-test was performed. I, 1N-MSE; II, 1N-SSIM; III, 3N-MSE; IV, 3N-SSIM

**Table 5 Comparison of quantitative OCTA feature analysis on en face images from conventional OCTA and different deep learning models.**

| Metric | GT | 1 N - MSE (I) | 1 N - SSIM (II) | 3 N - MSE (III) | 3 N - SSIM (IV) | GT vs I | GT vs II | GT vs III | GT vs IV |
|---|---|---|---|---|---|---|---|---|---|
| VAD | 0.3872 ± 0.0336 | 0.4223 ± 0.0118 | 0.3728 ± 0.0206 | 0.3785 ± 0.0223 | 0.3837 ± 0.0177 | 0.0713 | 0.3714 | 0.6375 | 0.8291 |
| VSD | 0.1501 ± 0.0123 | 0.2166 ± 0.0103 | 0.1865 ± 0.0146 | 0.1411 ± 0.0109 | 0.1414 ± 0.0067 | 0.0003 | 0.0064 | 0.2916 | 0.1556 |
| VPI | 0.2859 ± 0.0182 | 0.3503 ± 0.0142 | 0.3082 ± 0.0186 | 0.2666 ± 0.0154 | 0.2744 ± 0.0099 | 0.0014 | 0.0752 | 0.1279 | 0.2179 |

Values are reported as mean ± standard deviation.
For individual comparisons, a pair-wise two-way Student's *t*-test was performed.

**Connectivity analysis**. To assess disparities in vessel connectivity and overall vessel structure, we conducted a quantitative analysis of OCTA utilizing well-established metrics: vessel area density, vessel skeleton density, and vessel perimeter index. The outcomes of this analysis, which contrast the ground truths with deep learning predictions derived from models utilizing diverse inputs (1 N and 3 N) and loss functions (MSE and SSIM), are presented in Table 5.

Our examination reveals that the 1 N models consistently exhibit lower p-values in comparison to the ground truth, indicating detectable distinctions in values. Conversely, in the case of the 3 N models, we note both statistically insignificant disparities and relatively high *p*-values. This suggests that the predicted images maintain comparable structural connectivity to the ground truths. The evaluation of these quantitative metrics bears clinical relevance, as one of the fundamental applications of OCTA is the detection of retinal vascular changes.

**Retinopathy**. To assess the robustness of our proposed method, we conducted an evaluation using the best-performing model (3 N with SSIM loss) on an eye afflicted with proliferative diabetic retinopathy. Figure 8 presents representative images of this comparison. Notably, it becomes evident that the model's en face prediction effectively enhances the visualization of micro-aneurysms when compared to conventional OCTA. Additionally, upon evaluating the cross-sectional B-scans, we can observe a heightened brightness of vessels that exhibit low contrast in the conventional OCTA images, as depicted in the predicted image. An overarching consistency is observed in the blinking artifacts within the OCT en face image, which is markedly conspicuous in the conventional OCTA en face image. However, within the predicted en face image, a noticeable smoothing effect is apparent, yielding an overall improvement in image quality.

**Discussion**

In this study, we reported a fully automated convolutional network (FCN), SVC-Net, for OCTA construction that leverages the use of spatial vascular connectivity in OCT for vascular structure prediction using a single OCT volume. We quantitatively determined the optimal number of adjacent B-scans for the input into SVC-Net and the differences in the number of B-scans used for SV calculation between TVC and SVC. We conclude that three adjacent neighbors, 3 N, is the most optimal input into SVC-Net. We quantitatively compare the effects of SVC by comparing the performance of using two different inputs, a single OCT B-scan input, 1 N, and three adjacent OCT B-scan inputs, 3 N. We demonstrate that the 3 N model has superior performance compared to the 1 N model. In addition, we also compare the effects of different loss functions, i.e., MSE and SSIM loss functions, on the model's performance. Our study demonstrates that the SSIM loss function has superior performance over the MSE loss function. Our proposed method has been trained and tested on both animal and human OCT datasets. The ability to leverage single OCT volumes to generate OCTA can increase the speed of image acquisition by alleviating the need for multiple repetitions, reducing eye movement, and potentially increasing the FOV.

OCTA construction requires the acquisition of multiple OCT repetitions at the same imaging location, which, therefore, limits the imaging speed and FOV. In this study, we performed an ablation study to compare the effects of using different numbers of adjacent B-scans using SV calculation for OCTA construction. For quantitative comparison, the 2 N and 4 N had more noise compared to the 3 N. Qualitative observation, in particular for the human dataset, 4 N results in a pseudo vessel doubling artifact due to the larger area used for SV calculation. Therefore, 3 N had the optimal performance and was chosen as input into SVC-Net.

This observation has theoretical support in that the adjacent B-scans correspond to both spatial and temporal differences. Therefore, it carries information that can be used to estimate areas of hemodynamic changes, i.e., vascular tissue. For the SV calculation, the method uses a vector to determine the OCTA, i.e., for 3 N, it uses a vector of length 3. In principle, using an FCN, the model can leverage a localized region. For example, the standard convolutional filter size is of $3 \times 3 \times N$, as an input into the first layer of the SVC-Net, the FCN uses a localized region of

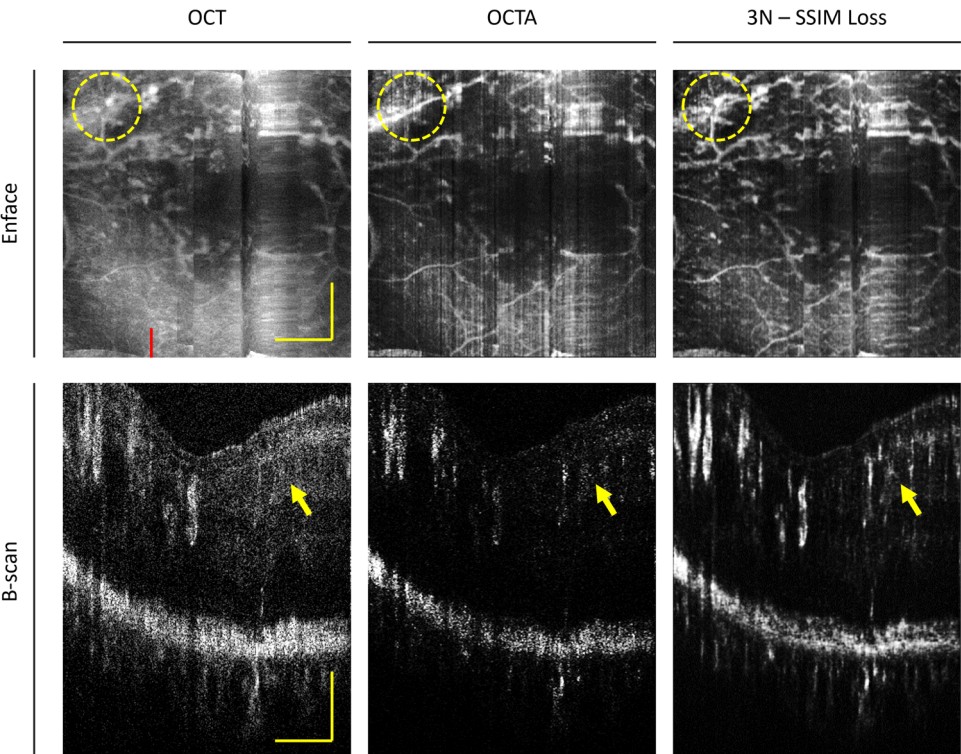

**Fig. 8 Comparative performance on retinopathy in the human eye.** Representative image of comparisons of en face projections from the inner limiting membrane to external limiting membrane for optical coherence tomography (OCT), conventional OCT angiography, and our proposed model with input three adjacent neighbors (3 N) optimized by the structural similarity index measure (SSIM) loss function on an eye with retinopathy. The yellow circle and arrows highlight areas of improved visualization of vascular abnormalities. The red line on the en face OCT denotes the B-scan location. The yellow scale bar in the en face image represents 600 μm. At the same time, the yellow scale bar in the B-scan represents 600 μm and 200 μm in the lateral and axial directions, respectively.

$3 \times 3 \times 3$. In addition, as the information is carried through the FCN, global information is also used in the decision-making process. Therefore, the FCN can better predict vascular tissue compared to the SV method due to the larger number of pixels it can leverage. The performance of SVC-Net using the 3 N model reveals improved vascular connectivity compared to the SV-3N method, supporting the hypothesis that the FCN is able to leverage a larger number of pixels for vessel prediction. On the human dataset, we do note that the smaller vessels in the SVP have less contrast compared to the DVP in the deep learning models. This may be due to the strong signal from the nerve fiber layer, which may minimize the signal for the smaller vessels in the SVP. In the DVP, the vessel structure between the different models, i.e., 1 N and 3 N, are similar because the DVP is bounded by two hypo-reflective layers, namely the inner nuclear layer and the outer nuclear layer. Therefore, the contributing signal for vascular prediction can be clearly determined by the CNN.

There have been a limited number of studies that have explored methods to alleviate this limitation using deep learning. Lee et al. demonstrated the single OCT B-scan input for OCTA construction in a human dataset using a similar U-Net type model[34]. In their work, they demonstrated that using an input-B-scan to output B-scan strategy, they can primarily predict the large blood vessels. At the same time, the smaller capillary-sized vessels have poor contrast and higher levels of noise. The results in this study for single OCT input are consistent with the results presented in Lee et al. This could primarily be due to the large vessels having better contrast compared to the smaller vessels in the OCT B-scan. In the study by Li et al., they demonstrate an input volume to output volume

strategy in animal models using a generative adversarial network[35]. Where the input is three adjacent OCT B-scans, and the output is three adjacent OCTA B-scans. While their results did not demonstrate capillary-level vessel structures, they did demonstrate that the use of SVC can help the deep learning model to predict higher performance metrics. The results in our study for SVC demonstrate capillary level vessels in both animal and human datasets. Overall, our methodology differs from the two aforementioned studies in that our strategy follows an input-volume-to-output B-scan strategy. The connectivity between the adjacent B-scans can provide the required information to accurately predict vessels of varied sizes.

In deep learning, there are many different hyperparameters that can be optimized for improved performance. Many studies often focus on the network architecture design, e.g., the depth or width of the network, or they develop different operations, e.g., atrous convolutions, depth-wise convolutions, etc. While all these hyperparameters play a role in the model's performance, one of the most fundamental hyperparameters of a CNN is the loss function layer. The choice of the loss function ultimately drives CNN's ability to learn its intended task[36]. In this study, we performed an ablation study to compare two loss functions, the MSE and SSIM. The results of our study, when compared to Lee et al.[34], using a single input and optimized with the MSE loss function on the animal and human dataset, demonstrate mainly large vessels are predicted, and the smaller vessels have poorer contrast. In our study, when we optimize the model using the SSIM loss function, we can observe a lower level of noise and improved vascular prediction for the animal dataset and human dataset. There are also quantifiable differences as measured using

the PSNR; we can observe an improved PSNR for both the SVP and DVP between the MSE and SSIM models.

Traditionally, MSE has been used for image reconstruction tasks. The MSE compares the ground truth and the predicted CNN image at the individual pixel level. The MSE models a quadratic function. Therefore, it can be easy to optimize due to its singular global minima characteristic. However, in many cases, an individual pixel is related to its surrounding pixel. In this case, there is a limitation to how much the MSE can optimize the deep learning model. On the other hand, the SSIM as a loss function evaluates three different parameters: luminance, contrast, and structure for a localized patch in the image. SSIM has been extensively used as an image quality metric. Therefore, it is reason that applying the SSIM as a loss function in image construction tasks can better optimize the deep learning model. When we combine the SVC and the loss function, we achieve the best-performing model. The model is trained to use the localized connectivity of the input, i.e., the adjacent B-scans, and is further optimized to achieve localized structural similarity in the predicted OCTA B-scan.

We have proposed a novel approach for OCTA construction using a single OCT volume for capillary-level visualization. However, there are some limitations with this study; for each of the dataset types (animal or human), the study is limited to a single OCT device. To demonstrate the generalization of this method, validation on different devices should be implemented. In addition, as an initial study, the dataset is limited to healthy eyes, in particular for human subjects. We evaluated our model on one disease name, namely proliferative diabetic retinopathy. For future considerations, we would need to evaluate this method in different eye conditions and disease states. Different eye conditions may affect the connectivity of the vasculature differently and, therefore, need to be further elucidated. Another variable to consider is that our proposed method has primarily been validated for OCTA constructed using the SV method; future studies should consider validating this method for other types of OCTA construction algorithms, e.g., OMAG and SSADA, as there may be performance differences as different construction algorithms rely on different information, e.g., phase or complex signal.

The SVC-Net for deep learning construction of micro-capillary resolution OCTA from single-scan-volumetric OCT has been developed and validated. A comparative study shows that the SVC in single-scan-volumetric OCT provides equivalent information to the TVC in multi-scan-volumetric OCT for robust OCTA construction. The SSIM loss function provides superior performance, compared to the MSE loss function, to optimize deep learning visualization of microstructures, such as microcapillaries, in single-scan-volumetric OCT. The combination of SVC involvement and SSIM loss function enabled robust OCTA construction from single-scan-volumetric OCT. With single-volumetric-scan OCT for rapid OCTA construction, the SVC-Net holds great promise to increase the imaging speed and thus enable rapid wide-field OCTA and dynamic monitoring of vascular changes to advance the clinical management of eye diseases.

## Methods

**Datasets**. *OCT volumetric data*. For the algorithm development, we optimized and evaluated SVC-Net on two dataset types, namely animal and human OCT datasets. For the animal dataset, 16 volumes comprised the dataset, 9 volumes for training, 1 volume for validation, and 6 volumes for testing. The total number of images used for training, validation, and testing were 5382, 598, and 3588 images, respectively. For testing en faces

from 6 volumes were used to evaluate model performance. For the human dataset, 16 volumes were used for training, 1 volume for validation and 5 volumes for testing. The total number of images used for training, validation, and testing were 4768, 298, and 1490 images, respectively. For testing en faces from 6 volumes were used to evaluate model performance. Additionally, one from a patient with retinopathy was used to demonstrate the model's qualitative performance for vascular abnormalities. For image acquisition, animal and human OCTs were taken with our custom lab-built OCT system. The system designs can be found in Supplementary Notes 1 and 2.

*OCTA construction*. The OCT scan pre-processing starts with registration of the OCT volume. The method that was employed for frame registration was the Discrete Fourier Transform registration method[37]. Since the OCT volume contains multiple repeated scans, the first step is to perform intra-frame registration, where each repetitive scan is registered to the first scan. This process is repeated for all scans. Next, inter-frame registration is performed to register each of the scans within the volume. After the OCT Volume Pre-processing, SVC input was generated using the inter-frame registered OCT B-scans. Meanwhile, to generate the ground truth, intensity-based SV processing was applied to intra-frame registered OCT B-scans using the method in[38]. SV processing algorithm can be found in Supplementary Note 3. Furthermore, for the ablation study to compare TVC and SVC, we used SV processing with varying numbers of repeated B-scans and varying numbers of adjacent neighbors, respectively. In this study, the number of adjacent numbers referred to as two, three, and four neighbors, referred to as 2 N, 3 N, and 4 N, respectively, was performed. To be consistent with the connotation, if only 1 B-scan was used, we refer to it as 1 N.

**Ethics declaration**. All animal experiments were approved by the local animal care and biosafety office and performed following the protocols approved by the Animal Care Committee (ACC) at the University of Illinois at Chicago (ACC Number: 19-044). This study followed the Association for Research in Vision and Ophthalmology Statement for the Use of Animals in Ophthalmic and Vision Research. All human experiments were approved by the Institutional Review Board of the University of Illinois at Chicago and were in pursuance of the ethical standards stated in the Declaration of Helsinki.

**Deep learning implementation**. *Deep learning model*. Our model, SVC-Net, was built using the methods described by Ahmed et al.[39], and the design is an encoder-decoder architecture, as shown in Fig. 9a. For the encoder, the EfficientNetB0 neural network[40] was employed. The decoder was designed using the Keras library, and the individual block components are illustrated in Fig. 9b. Briefly, we used a convolutional neural network to predict vessels in an image regression manner. The input into the CNN was a multichannel input comprised of OCT B-scans, and the output was a grayscale image. For other hyperparameters and training details, see Supplementary Note 4. The parameters of the CNN were optimized by training it on SVC inputs from single-scan OCT with the ground truth corresponding to OCTA images. The model was similarly trained on single channel inputs from OCT as well to determine the effects of SVC. To evaluate our model, we tested it on OCT volumes that were excluded from the training dataset.

*Loss function and evaluation metrics*. The loss layer of a neural network compares the output of the network with the ground truth. In this paper, we evaluate the effect of two loss functions, MSE and SSIM, on the performance of the model for OCTA

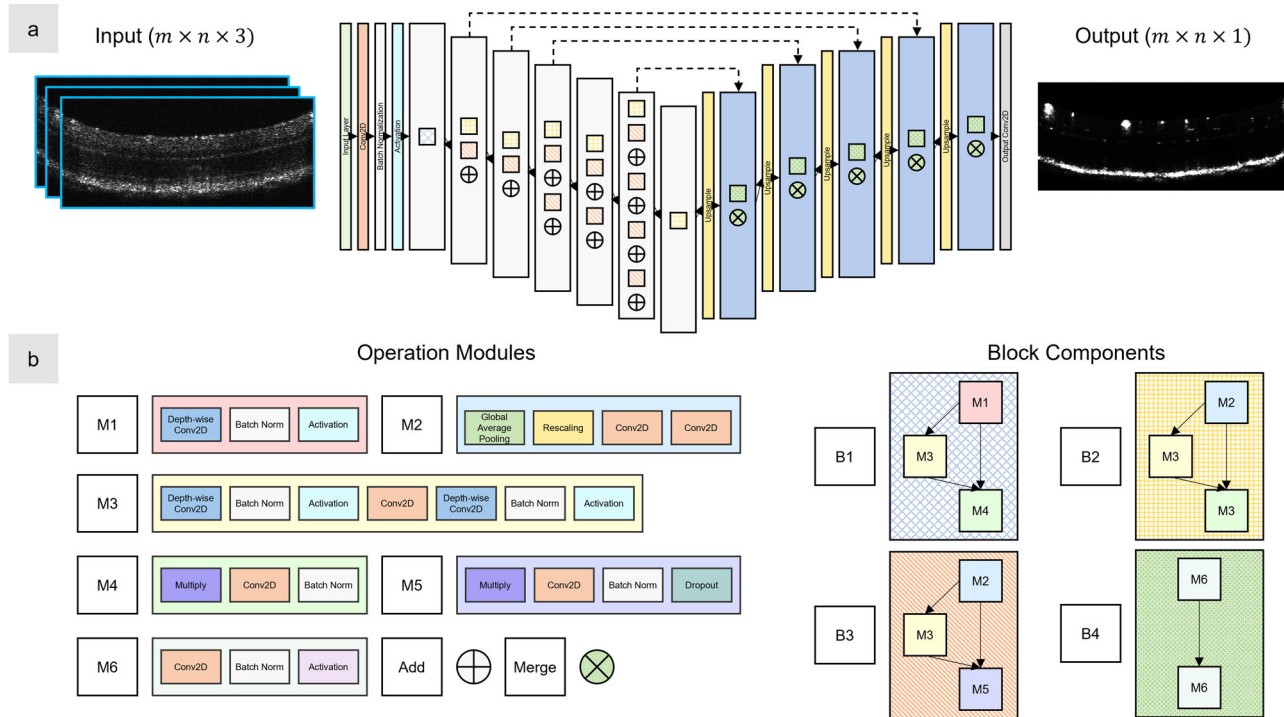

**Fig. 9 SVC-Net is based on an encoder-decoder architecture. a** The network architecture of SVC-Net, with representative OCT input images outlined in blue. The input is a three-channel image of size $m \times n \times 3$, and the output is a single-channel image of size $m \times n \times 1$. The network follows a U-shaped architecture with cross connections from the encoder to the decoder network represented by dashed arrows. For each layer, the information between each sub-block and operation flows from top to bottom. **b** The components of the network. The network is composed of modules M1 to M6. Each module contains a set of operations. The modules further make up the block components, which are represented as squares in the individual layers of (**a**).

construction. Therefore, in this section, we define the MSE and SSIM loss functions. For the formulation of MSE and SSIM, see Supplementary Note 5. To evaluate the performance of the model, we used PSNR and MS-SSIM. The evaluation metrics were applied to en face projections of SVP and DVP. Statistical methods included one-way analysis of variance (ANOVA) for multi-group comparisons, and post-hoc tests were conducted using pair-wise two-way Student's *t*-test. For the formulation of evaluation metrics and methodology of en face projection, see Supplementary Notes 6 and 7, respectively.

### Data availability

The data that support the findings of this study are available from the corresponding author upon reasonable request.

### Code availability

The code for the project is available on GitHub[41].

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

## Acknowledgements

This project was funded by the National Eye Institute (P30 EY001792, R01 EY023522, R01 EY030101, R01EY029673, and R01EY030842), Research to Prevent Blindness, and Richard and Loan Hill Endowment.

## Authors contributions

D.L. contributed to data preparation, network design, model implementation, data processing, statistical analysis, and paper preparation. T.S. contributed to data preparation, model implementation, data processing, and paper preparation. T.H.K. contributed to data acquisition and preparation. T.A. contributed to data acquisition, data preparation, and analysis tools. M.A. contributed to network design and model implementation. S.A. contributed to data acquisition, data preparation, and model implementation. A.R. contributed to data processing and analysis tools. B.E. contributed to network design and model implementation. A.D. contributed to network design and model implementation. G.M. contributed to data acquisition and preparation. J.I.L. contributed to data acquisition and preparation. X.Y. supervised the project and contributed to the study design and paper preparation. All authors reviewed and approved the paper.

## Competing interests

The authors declare no competing interests.
