## [Peer Review File · Communications Engineering]

Reviewers' comments:

Reviewer #1 (Remarks to the Author):

Le et al. proposed a spatially vascular connectivity network to construct microcapillaries resolution angiography. The author explored the optimal number of neighboring B-scans in a single-scan-volumetric OCT to input the network and compared the performance of results using the different loss functions. However, there are a lot of details that are missed and confusing in the manuscript. Please find my comments in the following:

1. Many abbreviations either lack a definition or are defined after using the full spelling. Please double check the manuscript. The errors include but are not limited to the following:
 - (1) In the discussion, NFL, INL, ONL were not defined.
 - (2) SV was used first in results but not defined.
 - (3) In caption of Figure 10, RVC-Net was not defined. Is it SVC-Net? Please carefully check the spelling of the words.
 - (4) The abbreviation 'MS-SSIM' should be spelled out in full on first use.
 - (5) SVP was not defined on first use. There was no DVP definition, but DCP definition.
2. The captions of all figures are very confusing. In captions of Figures 5 and 6, the author did not clarify the number of neighbors OCT B-scan for A3, B3. In the captions of Figures 7 and 8, the author did not describe images A4-B5. The captions missed a lot of details and were very confusing. The author should double-check all the captions of figures and give detailed and clear descriptions.
3. In the results, the subsection "SVC-Net a framework for deep learning-based OCTA construction" and "SVC can be a reliable signal source for deep learning" includes many descriptions for methods. These sections have disorganized structures.
4. In the methods, the author mentioned that they used an intensity-based speckle variance (SV) processing in the intra-frame registered OCT B-scans to generate the ground truth of the network. The author provided an equation of SV processing in the supplementary methods. However, the author did not describe the meaning of each letter represented. The author should add these details.
5. In the results, the author used visual observation to evaluate the noise and capillaries reconstruction in the generated OCTA en face. However, visual observation is a very subjective measurement, which is not convincing. An objective and quantitative method should be performed here. In figures 7 and 8, the author observed the ground truth and the predicted images to compare the effects of the different loss functions. They think the best model can reduce noise and improve contrast. How about connectivity? The author should provide quantitative supporting evidence for these metrics.
6. In the evaluation, the author used Peak-Signal-to-Noise Ratio (PSNR) and the multiscale structural-similar-index-measure (MS-SSIM) to evaluate the performance of results. In Figure 9, MS-SSIM and PSNR between the ground truth and generated human OCTA en face images of deep vascular plexus (DVP) are about 0.6 and 8, demonstrating the generated images have bad quality and are not very similar to the ground truth. These metrics may not demonstrate that the generated OCTA images are reliable.
7. The author only applied the proposed method to healthy eyes. How about the eyes with pathology, like diabetic retinopathy? Does the pathology affect the performance of the algorithm? Can the algorithm predict the correct pathologies?

Reviewer #2 (Remarks to the Author):

In this manuscript, the authors present a deep-learning-based OCTA construction from single-scan volumetric OCT. The idea is very interesting. However, I have the following four comments:

1. The method only uses single-scan volumetric OCT to generate OCTA image, which leads to a significant uncertainty due to the different degrees of tissue changes between adjacent B-scans. So, the method is not reliable.

2. The method uses deep learning to generate OCTA image. Are the results stable between two different trainings?

3. I think the author should compare the method with more famous OCTA algorithms and provide the results using indexes.

4. The training process of the method still needs OCTA images generated from multiple-scan volumetric OCT to supervise the CNN. Am I right?

5. In line 45 and 46, "A potential solution lies in the use of deep learning algorithms. In recent years, deep learning, a subset of machine learning and artificial intelligence (AI), is making strides in ophthalmic research." , you need to add some citations. Also, in line 48, "An example application is deep learning for AI screening of retinopathies." , you need to add some citations.

Reviewer #3 (Remarks to the Author):

The paper presents a very interesting approach for optical coherence tomography angiography (OCTA) that uses spatial variance in the structural images rather than temporal variance in conventional OCTA methods. The benefit would be to avoid repeated acquisition at the same location, to generate the same OCTA image with regular OCT scans. The manuscript is straightforward. Comments are below.

1. While it is conceivable to reason the spatial intensity variance is the contrast for learning OCTA, to contrast conventional temporal variance, it is not well supported by the presented data. I would argue that fundamental contrast may be just the reflectance intensity from vessels. This is evident by that 1N still produced reasonable OCTA on large vessel and less so in capillaries. The 3N/4N model performs better can be simply because more data was incorporated into the training. While this may not necessarily impact the results, it could be misleading.

2. Does it matter to use every other neighboring B-scan to do 3N-5N model, instead of immediate neighboring frames? This would provide some insight into what is the minimum scanning density required for the proposed approach. It would also be useful to evaluate whether indeed SVC is the fundamental contrast here. I would imagine there will be intensity difference on retina tissue as well among different B-scan frames. If so, what is the difference between vessel and retina tissue?

3. It was not clear to me if the network is supervised with conventional OCTA as ground truth or not at the beginning of the results section. Please clarify in describing Fig. 1.

4. I would suggest labeling the column and rows with simple text (e.g. 1N, 3N, SSIM, MSE), like a table, to improve the readability and to avoid labeling panner with mixed letter and numbers (e.g. A1), Fig. 2-8. Figure 7-8 captions have not description on 4, 5 (last two column).

5. All statistical bar plot should include descriptive statistical information, such as sample number, and significance test.

Subject: Response to review comments

Manuscript: COMMSENG-23-0015-T

Title: SVC-Net: A spatially vascular connectivity network for deep learning construction of microcapillary resolution angiography

Authors: David Le, Taeyoon Son, Tae-Hoon Kim, Tobiloba Adejumo, Mansour Abtahi, Shaiban Ahmed, Alfa Rossi, Behrouz Ebrahimi, Albert Dadzie, Guangying Ma, Jennifer I. Lim, and Xincheng Yao

We sincerely thank the editors and reviewers for their valuable comments and constructive suggestions on the manuscript. In this revision, we have addressed all the review comments. Following is a summary of our response to the reviewers' questions/concerns. For your convenience, we cite each of the review comments before our response initiated with "Response".

Reviewer #1 (Remarks to the Author):

Le et al. proposed a spatially vascular connectivity network to construct microcapillaries resolution angiography. The author explored the optimal number of neighboring B-scans in a single-scan-volumetric OCT to input the network and compared the performance of results using the different loss functions. However, there are a lot of details that are missed and confusing in the manuscript. Please find my comments in the following:

1. Many abbreviations either lack a definition or are defined after using the full spelling. Please double check the manuscript. The errors include but are not limited to the following:

Response: In the revision, we corrected the abbreviation errors in the manuscript.

(1) In the discussion, NFL, INL, ONL were not defined.

- On Page 13, line 278-281, we defined the abbreviations.

"This may be due to the strong signal from the nerve fiber layer (NFL) which may minimize the signal for the smaller vessels in the SVP. In the DVP, the vessel structure between the different models, i.e., 1N and 3N, are similar because the DVP is bounded by two hypo-reflective layers, namely the inner nuclear layer (INL) and outer nuclear layer (ONL)."

(2) SV was used first in results but not defined.

- On page 4, line 104-105, we define the abbreviation.

"For TVC-based speckle variance (SV) processing, visual observations show as the number of repeated B-scans increases, the noise is reduced as illustrated by Fig. 2 and Fig. 3."

(3) In caption of Figure 10, RVC-Net was not defined. Is it SVC-Net? Please carefully check the spelling of the words.

- We corrected the spelling in the caption of the revised Figure 9 (formerly Figure 10).

(4) The abbreviation 'MS-SSIM' should be spelled out in full on first use.

- On page 5, lines 119-120 we defined the abbreviation:

"We quantify the multi-scale structural similarity index measure (MS-SSIM) and the peak-signal-to-noise-ratio (PSNR) for the TVC 2N and 3N, and the SVC 2N, 3N and 4N en face images."

(5) SVP was not defined on first use. There was no DVP definition, but DCP definition.

- We corrected this error and revised to use DVP consistently throughout the manuscript. On page 6 line 138-140, we defined the abbreviation:

“The primary usage of OCTA is to observe the en face projections of the retinal vascular layers, therefore, we perform both qualitative and quantitative analyses for the en face projection of the superficial vascular plexus (SVP) and deep vascular plexus (DVP).”

2. The captions of all figures are very confusing. In captions of Figures 5 and 6, the author did not clarify the number of neighbors OCT B-scan for A3, B3. In the captions of Figures 7 and 8, the author did not describe images A4-B5. The captions missed a lot of details and were very confusing. The author should double-check all the captions of figures and give detailed and clear descriptions.

Response: We revised the figures and captions to improve readability. For example, in Figure 6, we labeled the columns and rows with simple texts, e.g., “OCTA”, “SVP”, “DVP” and updated the caption as follows:

“Fig. 6 Effect of loss function on en face OCTA prediction on mouse eye. Representative en face images from OCTA ground truth, and predictions using different loss functions (MSE and SSIM) and input images (1N and 3N) for SVP and DVP layers.”

3. In the results, the subsection “SVC-Net a framework for deep learning-based OCTA construction” and “SVC can be a reliable signal source for deep learning” includes many descriptions for methods. These sections have disorganized structures.

Response: In this revision, we simplified the section names to improve organization and conciseness. For example, we revised “SVC-Net a framework for deep learning-based OCTA construction” to “The deep learning framework” and “SVC can be a reliable signal source for deep learning” to “Optimization of neighboring scans”.

4. In the methods, the author mentioned that they used an intensity-based speckle variance (SV) processing in the intra-frame registered OCT B-scans to generate the ground truth of the network. The author provided an equation of SV processing in the supplementary methods. However, the author did not describe the meaning of each letter represented. The author should add these details.

Response: We updated the supplementary methods to include the description of each letter.

$$SV_{ij} = \frac{1}{N} \sum_i^N \left[I_{ijk}(x, z) - \frac{1}{N} \sum_i^N I_{ijk}(x, z) \right]^2 = \frac{1}{N} \sum_i^N [I_{ijk} - (I_{mean})_{jk}]^2 \quad (13)$$

Where i, j and k are indices of frame, lateral and depth pixel of the OCT B-scan, respectively. N is the number of frames used in the calculation. $(I_{mean})_{jk}$ is the averaged frame of N frames over the sample pixel.

5. In the results, the author used visual observation to evaluate the noise and capillaries reconstruction in the generated OCTA en face. However, visual observation is a very subjective measurement, which is not convincing. An objective and quantitative method should be performed here. In figures 7 and 8, the author observed the ground truth and the predicted images to compare the effects of the different loss functions. They think the best model can reduce noise and improve contrast. How about connectivity? The author should provide quantitative supporting evidence for these metrics.

Response: Thank you for the comment. To evaluate connectivity, we decided to perform quantitative OCTA analysis on the human dataset due to its clinical relevance. We evaluated three basic parameters, namely, vessel area density (VAD), vessel skeleton density (VSD), and vessel perimeter index (VPI) and compared the ground truth (derived from conventional OCTA) to our deep learning models, 1N-MSE, 1N-SSIM, 3N-MSE, and 3N-SSIM. The results of our analysis are included in the updated table 5. We added an additional section in results to discuss this analysis.

On Page 10 – 11, lines 215 – 231:

“Connectivity analysis

To assess disparities in vessel connectivity and overall vessel structure, we conducted a quantitative analysis of OCTA utilizing well-established metrics: vessel area density (VAD), vessel skeleton density (VSD), and vessel perimeter index (VPI). The outcomes of this analysis, which contrast the ground truths with deep learning predictions derived from models utilizing diverse inputs (1N and 3N) and loss functions (MSE and SSIM), are presented in Table 5.

Our examination reveals that the 1N models consistently exhibit lower p-values in comparison to the ground truth, indicating detectable distinctions in values. Conversely, in the case of the 3N models, we note both statistically insignificant disparities and relatively high p-values. This suggests that the predicted images maintain comparable structural connectivity to the ground truths. The evaluation of these quantitative metrics bears clinical relevance, as one of the fundamental applications of OCTA is the detection of retinal vascular changes.”

6. In the evaluation, the author used Peak-Signal-to-Noise Ratio (PSNR) and the multiscale structural-similar-index-measure (MS-SSIM) to evaluate the performance of results. In Figure 9, MS-SSIM and PSNR between the ground truth and generated human OCTA en face images of deep vascular plexus (DVP) are about 0.6 and 8, demonstrating the generated images have bad quality and are not very similar to the ground truth. These metrics may not demonstrate that the generated OCTA images are reliable.

Response: We want to clarify that the metric values of 0.6 and 8, MS-SSIM and PSNR respectively, are for the comparison of the conventional speckle variance using different scans, i.e., temporal repeated scans and spatial neighboring scans. We quantified the values for the different neighbors used to determine the optimal number of b-scans for the input into the deep learning model. Using the conventional speckle variance on neighbored B-scans to calculate OCTA results in insufficient images. Therefore, it motivated us to use a deep learning approach, which is adaptable and can learn from SVC input. In our proposed method, results, which we updated as table 3. For the human dataset, we demonstrate MS-SSIM and PSNR scores of 0.6-0.7 and 14-16, which we believe suggests adequate performance.

7. The author only applied the proposed method to healthy eyes. How about the eyes with pathology, like diabetic retinopathy? Does the pathology affect the performance of the algorithm? Can the algorithm predict the correct pathologies?

Response: To demonstrate the applicability of the algorithm on disease eye, we showcase an example of proliferative diabetic retinopathy in Fig. 8. The dataset was from a patient cohort of a collaborating clinician (J.I.L.). The algorithm demonstrates reliable performance on retinopathy, correctly predicting vessels, as well as vascular abnormalities, e.g., microaneurysm. We added a section in the results as follows:

On page 11 – 12, lines 232 – 246:

“Retinopathy

To assess the robustness of our proposed method, we conducted an evaluation using the best-performing model (3N with SSIM loss) on an eye afflicted with proliferative diabetic retinopathy (PDR). Fig. 8 presents representative images of this comparison. Notably, it becomes evident that the model's en face prediction effectively enhances the visualization of microaneurysms when compared to conventional OCTA. Additionally, upon evaluating the cross-sectional B-scans, we can observe a heightened brightness of vessels that exhibit low contrast in the conventional OCTA images, as depicted in the predicted image. An overarching consistency is observed in the blinking artifacts within the OCT en face images, which is markedly conspicuous in the conventional OCTA en face images. However, within the predicted en face images, a noticeable smoothing effect is apparent, yielding an overall improvement in image quality.”

We also added the patient recruitment information in our supplementary section:

“Additionally, three eyes from three patients with proliferative diabetic retinopathy (PDR) was imaged. Images that contained poor signal quality were excluded. Only one retinopathy eye was used to evaluate our model. The patient was recruited from the UIC Retinal Clinic. All recruited patients underwent complete anterior and dilated posterior segment examination by an experienced ophthalmologist (JIL). They also underwent contact lens examination with a slit lamp to identify PDR signs. The patients were classified based on the severity of DR (mild, moderate and severe NPDR, and PDR) according to the Early Treatment Diabetic Retinopathy Study (ETDRS) staging system. According to the American Academy of Ophthalmology, the ETDRS levels are classified as the following: no apparent retinopathy (Level 10), mild NPDR (Level 20), moderate NPDR (Level 35), severe NPDR (Level 53), and PDR (level 61) [38]. The patient was imaged using the custom OCT system and protocol.”

Reviewer #2 (Remarks to the Author):

In this manuscript, the authors present a deep-learning-based OCTA construction from single-scan volumetric OCT. The idea is very interesting. However, I have the following four comments:

1. The method only uses single-scan volumetric OCT to generate OCTA image, which lead to a significant uncertainty due to the different degrees of tissue changes between adjacent B-scans. So, the method is not reliable.

Response: We would like to clarify that the ground truth used for training the deep learning model is derived from conventional speckle variance OCTA calculation using four repeated b-scans. Whereas the training input is OCT B-scans derived from single-scan volumetric OCT. We added a sentence to clarify our ground truth generation.

Page 3, Lines 76-79: “The ground truth is based on conventional speckle variance OCTA construction from four repeated OCT B-scans. The input into SVC-Net will be comprised of OCT B-scans from single-scan OCT volume and the output of SVC-Net will be the OCTA B-scan.”

2. The method use deep learning way to generate OCTA image. Are the results stable between two different trainings?

Response: The results are overall stable between the training for both mouse and human datasets. As we employed data augmentation to increase the training set size and to minimize the effect of overfitting. We added a sentence to explain our use of data augmentation.

Page 21, Lines 558-559: “To promote stable training between different models and datasets, data augmentation, in the form of, i.e., horizontal flips, zoom, vertical and horizontal shifting, was implemented.”

3. I think the author should compare the method with more famous OCTA algorithm and provide the results using indexes.

Response: In the context of this paper, we focused on the proof-of-concept of using deep learning for OCTA generation. We plan in future studies to evaluate the effect and comparison of different OCTA algorithms with deep learning performance. We added a sentence in our discussion about future considerations for different OCTA algorithms.

Page 14, Lines 328-332: “Another variable to consider is that our proposed method has primarily been validated for OCTA constructed using the SV method, future studies should consider validating this method for other types of OCTA construction algorithms, e.g., OMAG and SSADA, as there may be performance differences as different construction algorithms rely on different information, e.g., phase or complex signal.”

4. The training process of the method still need OCTA images generated from multiple-scan volumetric OCT to supervise the CNN. Am I right?

Response: Yes, SVC-Net is trained using supervised learning. Therefore, the ground truth OCTA images were generated using speckle variance from multiple-scan volumetric OCT.

5. In line 45 and 46, “A potential solution lies in the use of deep learning algorithms. In recent years, deep learning, a subset of machine learning and artificial intelligence (AI), is making strides in ophthalmic research.”, you need to add some citations. Also, in line 48, “An example application is deep learning for AI screening of retinopathies.”, you need to add some citations.

Response: In the revision, we added citations to the following:

Page 2, Lines 45-46: “In recent years, deep learning, a subset of machine learning and artificial intelligence (AI), has been making strides in ophthalmic research [6-10].”

Page 2, Lines 48-49: “An example application is deep learning for AI screening of retinopathies [11-15].”

Reviewer #3 (Remarks to the Author):

The paper presents a very interesting approach for optical coherence tomography angiography (OCTA) that uses spatial variance in the structural images rather than temporal variance in conventional OCTA methods. The benefit would be to avoid repeated acquisition at the same location, to generate the same OCTA image with regular OCT scans. The manuscript is straightforward. Comments are below.

1. While it is conceivable to reason the spatial intensity variance is the contrast for learning OCTA, to contrast conventional temporal variance, it is not well supported by the presented data. I would argue that fundamental contrast may be just the reflectance intensity from vessels. This is evident by that 1N still produced reasonable OCTA on large vessel and less so in capillaries. The 3N/4N model performs better

can be simply because more data was incorporated into the training. While this may not necessarily impact the results, it could be misleading.

Response: Thank you for the comment. In this paper, we demonstrate using this deep learning approach is able to construct both large and capillaries vessels in animal and human OCT data. For example, in the mouse dataset, using the 1N input, the capillaries cannot be clearly constructed. Whereas the use of 3N input provided enough information for the deep learning model to construct the capillaries.

2. Does it matter to use every other neighboring B-scan to do 3N-5N model, instead of immediate neighboring frames? This would provide some insight into what is the minimum scanning density required for the proposed approach. It would also be useful to evaluate whether indeed SVC is the fundamental contrast here. I would imagine there will be intensity difference on retina tissue as well among different B-scan frames. If so, what is the difference between vessel and retina tissue?

Response: We believe the proximity of the scans does play a role in how well the model is able to reconstruct the vessels. In our analysis of the SVC, we observed that using four adjacent scans tends to blur the vessels as there's too much discontinuity. As proof of concept, we want to first evaluate the feasibility of using this neighboring frame approach. In the following study, we would like to further evaluate the minimum scanning density in the following study.

In terms of the differences between retinal and vessel tissue, there may be intensity differences on retina tissue among different B-scans. However, since the neighboring scan is used as input, in addition to intensity differences between vessel and retina tissue, there are also spatial/geometric differences, i.e., circularity of cross-sectional vessels, as compared to retinal tissue that the deep learning model can utilize for vessel prediction.

3. It was not clear to me if the network is supervised with conventional OCTA as ground truth or not at the beginning of the results section. Please clarify in describing Fig. 1.

Response: The ground truth was based on conventional OCTA construction, specifically using speckle variance with four repeated B-scans. We clarified the ground truth in the results section.

“The input into SVC-Net will be comprised of OCT B-scans from single-scan OCT volume and the output of SVC-Net will be the OCTA B-scan. The ground truth is based on conventional speckle variance OCTA construction from four repeated OCT B-scans.”

4. I would suggest labeling the column and rows with simple text (e.g. 1N, 3N, SSIM, MSE), like a table, to improve the readability and to avoid labeling panner with mixed letter and numbers (e.g. A1), Fig. 2-8. Figure 7-8 captions have not description on 4, 5 (last two column).

Response: We revised the figures to improve readability and to avoid labeling panner with mixed letter and numbers. For example, in Figure 6, we labeled the columns and rows with simple texts, e.g., “OCTA”, “SVP”, “DVP” and updated the captions accordingly.

5. All statistical bar plot should include descriptive statistical information, such as sample number, and significance test.

Response: We changed the bar plots into tables to provide descriptive statistical information, i.e., sample numbers, averages, standard deviations and significance tests.

REVIEWERS' COMMENTS:

Reviewer #3 (Remarks to the Author):

The authors reasonably addressed my previous comments.